# Metachromatic Leukodystrophy in Morocco: Identification of Causative Variants by Next-Generation Sequencing (NGS)

**DOI:** 10.3390/genes15121515

**Published:** 2024-11-26

**Authors:** Miloud Hammoud, María Domínguez-Ruiz, Imane Assiri, Daniel Rodrigues, Nisrine Aboussair, Val F. Lanza, Jesús Villarrubia, Cristóbal Colón, Naima Fdil, Francisco J. del Castillo

**Affiliations:** 1Metabolic Platform, Biochemistry Laboratory, Team for Childhood, Health and Development, Faculty of Medicine, Cadi Ayyad University, Marrakech B.P. 7010, Morocco; m.hammoud.ced@uca.ac.ma (M.H.); imane.assiri2020@gmail.com (I.A.); naima.fdil@uca.ac.ma (N.F.); 2Moroccan Association for Inherited Metabolic Diseases, Morocco; 3Servicio de Genética, Hospital Universitario Ramón y Cajal, IRYCIS, 28034 Madrid, Spain; mdominguezr@salud.madrid.org; 4Centro de Investigación Biomédica en Red de Enfermedades Raras (CIBERER), 28034 Madrid, Spain; 5Congenital Metabolic Diseases Unit, Department of Neonatology, University Clinical Hospital of Santiago de Compostela, Instituto de Investigación Sanitaria de Santiago (IDIS), European Reference Network for Hereditary Metabolic Disorders (MetabERN), 15706 Santiago de Compostela, Spain; daniel.caiola.candeias@sergas.es (D.R.); cristobal.colon.mejeras@sergas.es (C.C.); 6Centro de Investigación Biomédica en Red de Enfermedades Raras (CIBERER), 15706 Santiago de Compostela, Spain; 7Genetics Department, Clinical Research Center, Mohammed VI University Hospital, Faculty of Medicine and Pharmacy, Cadi Ayyad University, Marrakech-Principal B.P. 2360, Morocco; nisrine.aboussair@uca.ac.ma; 8UCA de Genómica Traslacional y Bioinformática (UCA-GTB), Hospital Universitario Ramón y Cajal, IRYCIS, 28034 Madrid, Spain; val.fernandez@salud.madrid.org; 9Centro de Investigación Biomédica en Red de Enfermedades Infecciosas (CIBERINFEC), 28034 Madrid, Spain; 10Servicio de Hematología, Hospital Universitario Ramón y Cajal, IRYCIS, 28034 Madrid, Spain; jesus.villarrubia@salud.madrid.org; 11CSUR de Enfermedades Metabólicas, European Reference Network for Hereditary Metabolic Disorders (MetabERN), Hospital Universitario Ramón y Cajal, IRYCIS, 28034 Madrid, Spain

**Keywords:** rare diseases, lysosomal diseases, NGS, metachromatic leukodystrophy, functional assays, phenotypic variation

## Abstract

(1) Background: Most rare disease patients endure long delays in obtaining a correct diagnosis, the so-called “diagnostic odyssey”, due to a combination of the rarity of their disorder and the lack of awareness of rare diseases among both primary care professionals and specialists. Next-generation sequencing (NGS) techniques that target genes underlying diverse phenotypic traits or groups of diseases are helping reduce these delays; (2) Methods: We used a combination of biochemical (thin-layer chromatography and high-performance liquid chromatography-tandem mass spectrometry), NGS (resequencing gene panels) and splicing assays to achieve a complete diagnosis of three patients with suspected metachromatic leukodystrophy, a neurologic lysosomal disorder; (3) Results: Affected individuals in each family were homozygotes for harmful variants in the *ARSA* gene, one of them novel (c.854+1dup, in family 1) and the other already described (c.640G>A, p.(Ala214Thr), in family 2). In addition, both affected individuals in family 2 were carriers of a known pathogenic variant in an additionallysosomal disease gene, *GNPTAB* (for mucolipidosis III). This additional variant may modify the clinical presentation by increasing lysosomal dysfunction. (4) Conclusions: We demonstrated the deleterious effect of the novel variant c.854+1dup on the splicing of *ARSA* transcripts. We also confirmed the involvement of variant c.640G>A in metachromatic leukodystrophy. Our results show the power of diagnostic approaches that combine deep phenotyping, NGS, and biochemical and functional techniques.

## 1. Introduction

Rare diseases (RD), due to their low incidence (≤1 in 2000), present specific challenges in terms of management. Beyond the limited medical and scientific knowledge available for each condition, such as its causes, mechanisms of pathogenicity, and natural history, one of the major issues is the lack of awareness of RDs, even among medical specialists who encounter some of the most common RDs in their field along with their typical symptoms and signs [1]. In fact, many primary care professionals report that they rarely or never encounter RD patients [2], which leads to very few cases being identified at the primary care level [3].

Despite comprehensive efforts to improve this situation [4], RD patients often endure what is commonly referred to as a “diagnostic odyssey”. During this prolonged period, the disease remains unrecognized by numerous clinicians, sometimes for years, until a correct diagnosis is finally made [5], resulting in significant personal and societal costs.

With the exception of a few rare cancers and infectious diseases, nearly all RD are genetic disorders. This is certainly the case for lysosomal disorders, a group of about 70 RD (combined incidence: 1 in 5000–5500) caused by pathogenic mutations in some fifty-odd genes encoding lysosomal proteins, such as hydrolases, transporters, receptors, enzyme activators, etc. Different types of mutations in some genes result in two distinct lysosomal conditions (e.g., *GLB1* underlies both GM1 gangliosidosis and mucopolysaccharidosis type IV B) [6]. Unlike most rare diseases, 12 lysosomal disorders already have effective treatments that are routinely used in clinical practice, with more expected soon [7]. We classified lysosomal disorders into six different phenotypic groups according to the major symptoms or signs at onset [our unpublished data]. One of these groups encompasses neurological symptoms and signs, for which specialists must consider hundreds of differential diagnoses depending on the specific abnormalities observed.

The recent application of genomic techniques based on next-generation sequencing (NGS) has revolutionized the diagnosis of RD, making the goal of achieving a diagnosis for every RD patient both attainable and challenging [8]. Indeed, the use of NGS-based diagnostic tools eliminates the need for aclinical suspicion, enabling the screening of large numbers of genes that are involved in common phenotypic traits. We have shown the effectiveness of NGS-based gene panel resequencing for the rapid screening of suspected lysosomal disease cases, achieving significantly higher diagnostic success rates compared to the standard diagnostic approach [9]. Such a standard approach relies on biochemical analyses to identify specific enzyme deficiencies suggested by clinical presentation, followed by Sanger sequencing of the gene(s) involved. Here, we apply NGS-based gene panel techniques for the genetic diagnosis of children with suspected metachromatic leukodystrophy (MLD) [10], one of the lysosomal disorders within the neurological phenotypic group.

## 2. Materials and Methods

### 2.1. Subjects

The study was approved by the Ethical Committee of the Hospital Universitario Ramón y Cajal (in accordance with the 1964 Declaration of Helsinki) with approval code 249-23. We obtained written informed consent from all participating subjects.

Clinical examination of the subjects was part of routine clinical care. Clinical data were obtained from the subjects’ medical histories and through reexamination after a genetic diagnosis was established.

### 2.2. Biochemical Analyses

Biochemical screening for sphingolipidoses in samples from Moroccan patients was performed at the Metabolic Platform of the Biochemistry Laboratory of Cadi Ayyad University, as previously described [11]. Briefly, first-void morning urine samples were analyzed by thin-layer chromatography (TLC) to detect non-polar urinary lipids (including sulfatides, glycolypids, and sphingolipids). Subsequently, arylsulfatase A activity was determined colorimetrically, either on serum samples obtained from peripheral blood or on dried bloodspots (DBS), as previously described [12,13].

Additional biochemical experiments were performed at the Congenital Metabolic Diseases Unit of the University Clinical Hospital of Santiago de Compostela, where the MLD biochemical diagnosis on DBS is based on a two-tier strategy: quantification of sulfatides followed by arylsulfatase A enzymatic assay, both carried out by high-performance liquid chromatography-tandem mass spectrometry (HPLC-MS/MS).

For the quantification of the two most abundant sulfatides in DBS [14,15] (C16:0 and C16:0-OH), a 3.2 mm punch was extracted with 300 µL of methanol [15] containing 10 nM internal standard (d5-C16:0-sulfatide; Gelbchem, Seattle, WA, USA) in a 96 deep-well plate (Thermo Fisher Scientific, Waltham, MA, USA). As a control, 100 µL methanol containing 14.4 nM C16:0-sulfatide (external calibrator; Cayman Chemical Company, Ann Arbor, MI, USA) was included in triplicate and processed at the same time as the DBS. After 4 h of extraction at 37 °C with agitation, the plate was centrifuged for 5 min at 2200× *g*, and the supernatant was analyzed by HPLC-MS/MS in a Sciex 4500 MD QTRAP. Both sulfatides (transitions 778.6>96.8 and 794.5>96.8) and the internal standard (783.5>96.8) were detected by multiple reaction monitoring (MRM) in the electrospray ionization (ESI) negative mode. Separation was achieved with a Zorbax SB-C18 RRHT column (2.1 × 50 mm, 1.8 µm; Agilent Technologies, Santa Clara, CA, USA) at 45 °C, by using solvent A (50/50 water/acetonitrile with 0.1% formic acid) and solvent B (80/20 isopropanol/acetonitrile with 0.1% formic acid) and the following gradient: 0.0 min, 50% B, 0.25 mL/min; 1.00 min, 95% B, 0.25 mL/min; 1.01 min, 95% B, 0.35 mL/min, 1.40 min, 100% B, 0.35 mL/min; 1.41–1.65 min, 50% B, 0.35 mL/min; 1.66–1.80 min, 50% B, 0.25 mL/min.

For the arylsulfatase A enzymatic assay [16], a 3.2 mm DBS punch was eluted for 4 h in a 96-well plate (Greiner, 655101) with 50 µL of a solution of 0.23% NH_4_OH in assay buffer (80 mM sodium acetate buffer, containing 2 g/L sodium taurodeoxycholate, pH = 4.5). Meanwhile, Sephadex-G25 resin (Gelbchem, Seattle, WA, USA) (60 mg/assay) was swollen in milli-Q water (10 mL/g of dry resin) for 3 h with agitation. Afterward, 600 µL of resin slurry per well was transferred into a fritted deep-well plate (Thermo Fisher Scientific, Waltham, MA, USA) and layered on top of a collector plate (1000 µL 96-well plate). The plate was sealed and centrifuged for 1 min at 800× *g*. The solution in the collector plate was discarded, and 600 µL of assay buffer was added to the fritted plate and centrifuged at 800× *g* for 1 min. This process was repeated 2 more times. The collector plate was changed to a 500 µL 96 deep-well plate (Thermo Fisher Scientific, Waltham, MA, USA), and 30 µL of DBS extract was placed onto the resin, followed by 15 µL of assay buffer. The plate was centrifuged for 1 min at 800× *g*, and 10 µL of assay cocktail, prepared by adding 500 µL of assay buffer to a vial of reagent (Gelbchem, Seattle, WA, USA), was added to the eluate. The plate was sealed, centrifuged for 1 min at 2200× *g*, and incubated for 17 h at 37 °C with agitation. The reaction was quenched with 300 µL of methanol, and the plate was centrifuged a final time for 5 min at 2200× *g*. A 150 µL aliquot of the supernatant was transferred to a 96-well plate (Greiner, Kremsmünster, Austria), and 50 µL of milli-Q water was added to each well. The plate was sealed with aluminum foil, and the activity was measured by HPLC-MS/MS. Arylsulfatase A product (731.5>264.2) and internal standard (735.6>264.2) transitions were detected by MRM in the ESI-positive mode. Peak detection was achieved with an Acquity UPLC CSH Fluoro Phenyl column (2.1 × 50 mm, 1.7 µm; Waters Corp., Milford, MA, USA) at 45 °C, using solvent A (50/50 water/acetonitrile with 0.1% formic acid) and solvent B (50/50 isopropanol/acetonitrile with 0.1% formic acid) with the following gradient: 0.0–0.75 min, 0.0% B, 0.45 mL/min; 1.50 min, 90% B, 0.45 mL/min, 2.00 min, 100% B, 0.45 mL/min; 2.00–3.20 min, 100% B, 0.45 mL/min; 3.21–4.50 min, 0.0% B, 0.45 mL/min.

### 2.3. Genetic Techniques

DNA was extracted from peripheral blood samples using a FlexiGene DNA kit (Qiagen, Hilden, Germany) at the Core Research Genomic Facility (UCA-GTB) of Hospital Ramón y Cajal-IRYCIS.

Targeted massively parallel DNA sequencing was carried out using the Splenomegaly and Thrombocytopenia (STP) v3 gene panel, which was developed in our laboratory at Servicio de Genética of Hospital Ramón y Cajal-IRYCIS. STP v3 covers 498,923 bp of genomic sequence; it contains 101 genes (Appendix A) that are known to be involved in lysosomal diseases (including *ARSA*, the gene encoding arylsulfatase A and *PSAP*, the gene encoding the arylsulfatase A activator saposin B, both involved in different forms of MLD) and their differential diagnoses, and it is based on the SureSelect XT probe capture system [9,17] (Agilent Technologies, Santa Clara, CA, USA). The captured enriched libraries were sequenced on the Illumina MiSeq platform (Illumina, Inc., San Diego, CA, USA). Sequence data were mapped against the human genome GRCh37/hg19 reference sequence. Validation runs performed on a trio from CEPH family 1463, consisting of samples NA12891 (father), NA12892 (mother), and NA12878 (daughter), showed 99.67% sensitivity, 99.99% specificity, and 98.22% predictive positive value. The median depth across all regions was 411X, thereby complying with the laboratory standards for NGS published by the American College of Medical Genetics [18]. For all analyses, we reproduced the conditions used in the validation runs.

We used Qiagen Clinical Insight Interpret Translational (Qiagen, Hilden, Germany) to annotate, filter, classify, and prioritize single nucleotide variants and copy number variants. We successively filtered for allelic fraction (AF) (variants with AF < 20% are excluded), population frequency (PF) in the gnomAD and 1000 Genome databases (variants with PF ≥ 1% were excluded, whether in the whole database or in specific populations), functional consequences (variants are only retained if they are CNV, nonsense, missense, loss of start or of stop codons, frameshifts or in-phase insertions or deletions, affecting the 5 intronic nucleotides closest to the exon-intron boundary, or PHRED-scaled CADD values ≥ 15), and ACMG classification (only variants that are classified as pathogenic, likely pathogenic, or of unknown significance are retained), as applied by Qiagen Clinical Insight Interpret Translational.

For validation, we selected those variants that passed all the aforementioned filters, and that affected a gene involved in a known neurologic phenotype. We used Sanger DNA sequencing to confirm such variants and to study their segregation in the pedigrees. Variants that did not segregate with the disorder in the family were rejected. Primers and PCR conditions for amplicons with such variants are shown in Table 1. Sanger DNA sequencing was performed using an ABI Prism 3100 Avant Genetic Analyzer (Applied Biosystems, Waltham, MA, USA).

The microsatellite markers D22S1169, D22S1056, and (TG)_n_195 were amplified using fluorescently labeled primers and PCR conditions, as shown in Table 2. Amplified alleles were resolved by capillary electrophoresis using an ABI Prism 3100 Avant Genetic Analyzer (Applied Biosystems, Waltham, MA, USA).

### 2.4. Assessment of Pathogenicity of DNA Variants

Pathogenicity of DNA variants was assessed according to the guidelines of the American College of Medical Genetics and Genomics and the Association for Molecular Pathology (ACMG/AMP) [19], as implemented by both Qiagen Clinical Insight Interpret Translational and VarSome [20], using GRCh37/hg19 as the human reference genome.

### 2.5. Assay for Effects on Splicing of ARSA Variant c.854+1dup

We obtained new blood samples from individuals PET171 and PET172, heterozygous for the c.854+1dup variant. We extracted total RNA from these peripheral blood samples, preserved in PAXgene™ blood RNA tubes (PreAnalytiX, Hombrechtikon, Switzerland), using the PAXgene Blood RNA kit (Qiagen & BD, Venlo, The Netherlands, and Franklin Lakes, NJ, USA, respectively), following the manufacturer’s protocol. RNA was reverse-transcribed into cDNA using Superscript II enzyme (Invitrogen, Waltham, MA, USA) and random hexamer primers. To analyze the splicing of intron 4, we designed primers enabling amplification from exon 4 to exon 5 of the *ARSA* gene (Table 1), with the upper primer labeled with a fluorochrome. PCR was performed on cDNAs from the mutation carriers and two unrelated wild-type controls. Labeled PCR products were treated with T4 DNA polymerase (ThermoFisher, Waltham, MA, USA) to blunt any 3′ protruding termini, allowing for correct sizing and quantification of the product using an ABI Prism 3100 Avant Genetic Analyzer (Applied Biosystems, Waltham, MA, USA). We estimated the amount of each fluorescent amplicon as the area under the curve. In parallel, we also carried out the PCR with unlabeled primers to verify the sequence of the amplicons using the Sanger method.

## 3. Results

Seventy-three patients (age range: 18 days–59 years; median age 4 years) with symptoms and signs suggesting asphingolipidosis were referred to the Metabolic Platform of Cadi Ayyad University for biochemical screening of suspected metabolic diseases [11]. Patients in this cohort were characterized by psychomotor regression, central nervous system motor signs (pyramidal, cerebellar, dystonia), and other neurological symptoms (such as epilepsy, seizures, etc.). Magnetic resonance imaging (MRI) was performed in 43 of 73 patients, revealing pathological characteristics, including white matter anomalies in 32 subjects (74%). In three patients belonging to two consanguineous families (Figure 1), we carried out thin-layer chromatography analysis of urine samples and identified urinary excretion of sulfatides, suggesting a deficit of arylsulfatase A activity. Colorimetric determination of arylsulfatase A activity in plasma verified the deficit, suggesting late-infantile MLD in all three patients to be formally confirmed by genetic testing, as recommended by current guidelines [21].

### 3.1. Clinical Data

Patient PET170 in family 1 was the offspring of first-degree consanguineous parents (Figure 1). She began psychomotor regression at age 18 months that was clinically ascertained at age 2 years (Appendix A), with hypotonia, peripheral neuropathy, and ataxia. MRI performed at 5 years of age showed: (1) impaired white matter in T2 and FLAIR hypersignals, with a tigroid pattern and diffuse cortical and subcortical atrophy; (2) hypoplasia of the corpus callosum; (3) moderate hydrocephalus without obstruction affecting all four ventricles; (4) hypoplasia of the cerebellar vermis with thickened and maloriented superior cerebellar peduncles (molar tooth sign), reminiscent of Joubert syndrome; and (5) diffuse anomalies in the cerebellar hemispheres in T2 and FLAIR hypersignals. Arylsulfatase A activity in plasma was 15.21 nmol/mL/4 h (normal reference values: 35.16–177.12). She had two younger sisters (PET171 and PET172) who displayed some psychomotor symptoms and signs; however, because they were not referred to a clinician for examination, we assigned them an uncertain clinical status. Notably, thin-layer chromatography analysis detected traces of sulfatides in the urine of these two younger siblings.

Patients PET173 and PET174 in family 2 were the offspring of second-degree consanguineous parents (Figure 1). Both children began psychomotor regression at 30 and 36 months of age (Appendix A). After MRI, PET173 showed T2 and FLAIR hypersignals involving the periventricular white matter and the semioval center in confluent, bilateral, and symmetrical patches with a tigroid pattern, sparing U-shaped association fibers; diffuse T2 hypersignals affected the corpus callosum as well. No anomalies were detected in the ventricular system, cerebellum, cerebellopontine angle, or brainstem. PET174 displayed bilateral, confluent, and symmetrical demyelinating lesions in the periventricular white matter, sparing U-shaped association fibers. Their plasma enzyme activity levels were 26.25 (PET173) and 15.42 (PET174) nmol/mL/4 h.

### 3.2. Genetic Analysis by NGS

To identify the genetic variants underlying the MLD identified in the patients, we performed massively parallel targeted sequencing on the propositi of each family (PET170 and PET173) using the resequencing panel STP v3. STP v3 is based on liquid capture technology, and this panel targets all coding exons, all 5’-untranslated region exons, all exon-intron boundaries of each of the 101 genes included, plus any additional sequences, such as deep intronic sites or promoters, in which pathogenic mutations have been reported in the ClinVar database (https://www.ncbi.nlm.nih.gov/clinvar/, last accessed on 30 October 2018) [22]. The panel targeted 1821 genomic regions from 101 genes and 174 different Online Mendelian Inheritance in Man (OMIM) [23] phenotypes for a total 499 kb of sequence.

For family 1, our panel analyses showed that individual PET170 was homozygous (depth: 242 reads, AF: 100%) for duplication of a guanine nucleotide at the 5’ end of intron 4 of the *ARSA* gene (transcript NM_000487.6, variant c.854+1dup). We did not identify any other harmful variant in any of the genes analyzed by the panel, including *PSAP*. c.854+1dup was predicted by all bioinformatic analysis suites that we used (NNSplice 0.9 [24], MaxEntScan [25] and MutationTaster2021 [26]) to move the donor splice site of intron 4 downstream just by a single nucleotide, which would result in a frameshift and the eventual appearance of a premature termination codon, p.(Pro286Thrfs*2). To our knowledge, this variant has never been reported previously and is absent from the gnomAD [27] and 1000 Genomes [28] databases. Assessment of the variant using the ACMG/AMP pathogenicity guidelines classified c.854+1dup as **pathogenic** (with criteria **PVS1**, **PM2,** and **PP3**) (Table 3).

We confirmed both the existence of the variant and its segregation within the family by Sanger sequencing of *ARSA* exon 4 and adjacent intronic sequences (Figure 2). Results from haplotype analysis with closely linked microsatellite markers flanking the *ARSA* gene support that the same pathogenic allele was indeed inherited from both parents.

Interestingly, genetic analysis showed that the two sisters with uncertain clinical status (PET171 and PET172) were heterozygous for c.854+1dup, having inherited the same wild-type *ARSA* allele from their mother. We verified that this maternal allele did not harbor any harmful variants or the arylsulfatase A pseudodeficiency allele c.[1055A>G;*96A>G] [32]. Given that thin-layer chromatography analysis detected traces of sulfatides in the urine of both sisters, we reanalyzed their DBS using HPLC-MS/MS. The concentrations of C16:0 and C16:0-OH sulfatides and the activity of arylsulfatase A in DBS from both PET171 and PET172 were within normal reference values, confirming the wild-type status of these two subjects for arylsulfatase A activity.

As regards family 2, panel analysis showed that affected individual PET173 was homozygous (depth: 336 reads, AF: 100%) for missense variant c.640G>A, p.(Ala214Thr) (*ARSA* transcript NM_000487.6). Bioinformatic analyses provided conflicting evaluations of the pathogenicity of the c.640G>A variant. Application of ACMG/AMP pathogenicity guidelines classified the c.640G>A as **likely pathogenic** (with criteria **PM1**, **PM2,** and **PM5**), notably because another pathogenic missense variant, p.(Ala214Pro) affects the same amino acid residue (Clinvar database RCV001380269.7) (Table 3). Moreover, the c.640G>A, p.(Ala214Thr) variant had already been reported in compound heterozygosity with the pathogenic variant c.893G>T, p.(Gly298Val) as underlying MLD in a Chinese subject [29]. Sanger sequencing confirmed that both affected siblings, PET173 and PET174, were homozygous for c.640G>A; microsatellite analysis confirmed the segregation of the variant and the inheritance of the pathogenic allele from both parents (Figure 3).

To our surprise, in addition to being a homozygote for missense variant c.640G>A in the *ARSA* gene, PET173 was also heterozygous for two other known pathogenic variants in the genes *GNPTAB* (variant c.1931_1932inv, p.(Thr644Met), at transcript NM_024312.5; depth: 916 reads, AF: 49%) [30] and *MANBA* (variant c.1922G>A, p.(Arg641His) at transcript NM_005908.4; depth: 1680, AF: 48%) [31] (Table 3). These two variants underlie other distinct lysosomal disorders, mucolipidosis III and β-mannosidosis, both with autosomal recessive inheritance. We verified the carrier status of PET173 for the two variants by Sanger sequencing of exon 13 of *GNPTAB* and exon 14 of *MANBA*. Likewise, we determined that PET174 was a heterozygous carrier only of the variant c.1931_1932inv in *GNPTAB*.

Appendix A summarizes all our clinical, biochemical, and genetic findings.

### 3.3. Functional Analysis of ARSA Variant c.854+1dup

To verify whether variant c.854+1dup had effects on splicing, we analyzed total RNA from blood samples of heterozygous siblings PET171 and PET172. We detected two fluorescently labeled amplification products of *ARSA* transcripts, the first with the expected length of the wild-type allele and the second with just an additional nucleotide, putatively corresponding to the c.854+1dup allele (Figure 4). This second product was absent from the homozygous wild-type controls. As predicted, the consequence of the addition of this nucleotide is a shift in the reading frame of the mutant transcript and the appearance of a premature termination codon, as verified by Sanger sequencing of the unlabeled amplification products. Apparently, the mutant transcript largely escapes degradation by nonsense-mediated decay, as we observed sizable amounts of the mutant transcript (at about 60% of the level of the wild-type transcript). Hence, it appears that the c.854+1dup variant may indeed result in the synthesis of a truncated arylsulfatase A.

## 4. Discussion

Here, we combined diverse analytical techniques (biochemical, NGS, and molecular genetics) to achieve a complete diagnosis of MLD and to identify the first genetic variants underlying this disorder in Moroccan patients. Demonstrating low arylsulfatase activity is not sufficient for the diagnosis of MLD because of pseudodeficiency alleles [32] and variations in enzyme activity among carriers of pathogenic variants [33]. Therefore, current guidelines stress the importance of a combined clinical, biochemical, and genetic approach for diagnosis [21].

In family 1, we identified a novel homozygous pathogenic variant, c. 854 +1 dup in the *ARSA* gene of the proband, leading to late-infantile MLD. We showed the deleterious effects of this variant on the splicing of *ARSA* transcripts by performing a functional assay directly on total RNA obtained from the patient. This type of splicing analysis, unlike minigene splicing assays, reveals the actual cellular impact of the patient variant on *ARSA* transcription. Its limitation is a failure to amplify the mutant allele, which may be ascribed to diverse causes. However, in our case, the assay indicated that c.854+1dup likely results in the truncation of arylsulfatase A. This is in agreement with established genotype-phenotype correlations, in which protein-truncating variants in *ARSA* underlie the most severe form of the disorder, with late-infantile presentation [34]. We further showed that the propositus’ siblings, with uncertain clinical status, were not afflicted by MLD, indicating that their comparatively mild neurological symptoms and signs may be due to a different disorder that requires further clinical delineation.

In family 2, we identified that both patients were homozygous for the missense *ARSA* variant c.640G>A, p.(Ala214Thr). This variant has been previously described in a late-infantile Chinese patient who was compound heterozygous for c.640G>A and another missense variant [29] and in several other cases with an undisclosed phenotype and accompanying variant (ClinVar records RCV000177069.5 and RCV002517708.5). In family 2, clinical presentations and age of onset (30 and 36 months) were quite homogeneous between the two patients and seemed comparable to those of the Chinese subject (28 months). Both patients (homozygous for a missense variant) showed a slightly milder phenotype than the patient from family 1 (homozygous for a truncating variant), without any cerebellar involvement and with the later appearance of psychomotor regression.

However, genotype-phenotype correlations involving missense *ARSA* variants remain elusive. Indeed, patients harboring two missense or one missense and one protein-truncating variant in *ARSA* display variable clinical presentations (from severe to mild) and ages at onset (from late infantile to adult); moreover, intrafamilial variations in phenotype and age of onset are common [34]. The causes of these variations remain unknown. In this respect, the fact that both affected siblings are carriers of another known pathogenic variant of *GNPTAB* must be taken into account. *GNPTAB* is involved in mucolipidosis II and III, two lysosomal disorders with neurologic symptoms [35], and harmful variants in *GNPTAB* have been identified both as a risk factor for the development of Parkinson’s disease [36] and as enhancers of α-synuclein neurotoxicity in *Drosophila* [37]. Recent evidence supports the hypothesis that a long-term subclinical malfunction of the lysosome-autophagy machinery, caused by heterozygosity for certain variants in specific lysosomal disorder genes, will prevent the clearing of α-synuclein aggregates, eventually leading to neurodegeneration typical of Parkinson’s disease [38,39]. Thus, being a carrier for the pathogenic *GNPTAB* variant c.1931_1932inv may be a factor leading to increased severity of neurologic symptoms and/or earlier onset in the two patients of family 2. Since lysosomal dysfunction is a cause of neurodegenerative disease [40], it may be hypothesized that heterozygous pathogenic variants in other genes involved in lysosomal disorders may be one of the sources of variation in clinical presentations and age at onset among patients with MLD. This hypothesis deserves to be explored by additional genetic testing in existing MLD cohorts, in particular among affected relatives with intrafamilial phenotypic variation.

We have previously shown that among lysosomal disorder patients, being a carrier of pathogenic variants in additional lysosomal disorder genes distinct from the gene underlying their particular lysosomal disease is not uncommon [9]. This possible source of phenotypic variation is seldom considered because the currently accepted diagnostic strategy for lysosomal disorders emphasizes directing second-tier genetic testing only to the gene(s) underlying the enzymatic deficit identified in biochemical screening. Widespread application of NGS screenings, whether as first-tier or second-tier analysis of patients and paying attention to any carrier status of pathogenic/likely pathogenic variants in other lysosomal disorder genes, will help explore this hypothesis and may help perfect genotype-phenotype correlations. Furthermore, particular care should be devoted to analyzing the NGS data of patients from consanguineous marriages. Recent data indicate that the concurrence of two genetic disorders in a single patient or family is not as rare as previously thought [41]. Consanguinity will increase the probability of such concurrence; thus, consanguineous families with patients displaying unusual phenotypes warrant additional genetic analyses and clinical characterization/deep phenotyping (as in the case of individuals PET171 and PET172 from family 1).

In the cohort that we investigated, 47 of 73 subjects (64%) were the offspring of consanguineous unions [11], a proportion that is twice the frequency of this kind of marriage in the Southern Moroccan population. This underscores the high burden of disease associated with such customs, amplified by multiple loops of consanguinity in family ancestors. In this respect, early, accurate, and cost-efficient diagnosis is essential both to apply treatments as they gradually become available, and to provide preventive measures, such as genetic counseling, highlighting the need to increase early screening efforts. Indeed, in populations with a high percentage of consanguineous marriages, genetic counseling after the birth of the first affected child is essential to mitigate the elevated familial and societal burden of additional affected children (as in the case of family 2). The combination of relatively inexpensive biochemical analyses [11] and international research collaborations that provide access to cutting-edge genomic technology and help train personnel to implement and develop such techniques locally offers the best opportunity to face the challenge of timely, cost-effective metabolic disease screening in the populations of developing countries.

## Figures and Tables

**Figure 1 genes-15-01515-f001:**
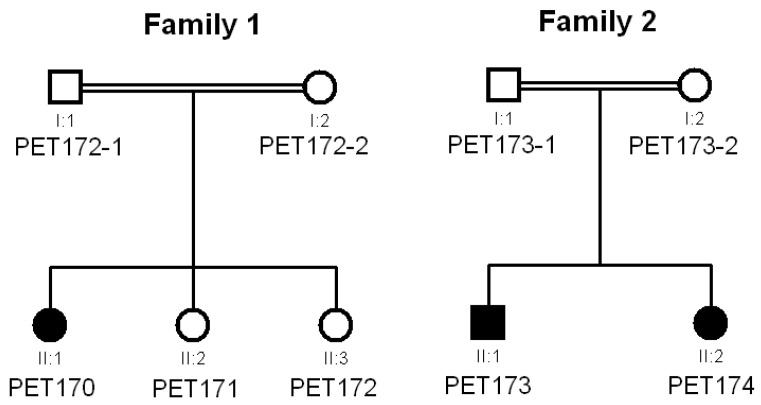
Pedigrees of the two families analyzed in this study.

**Figure 2 genes-15-01515-f002:**
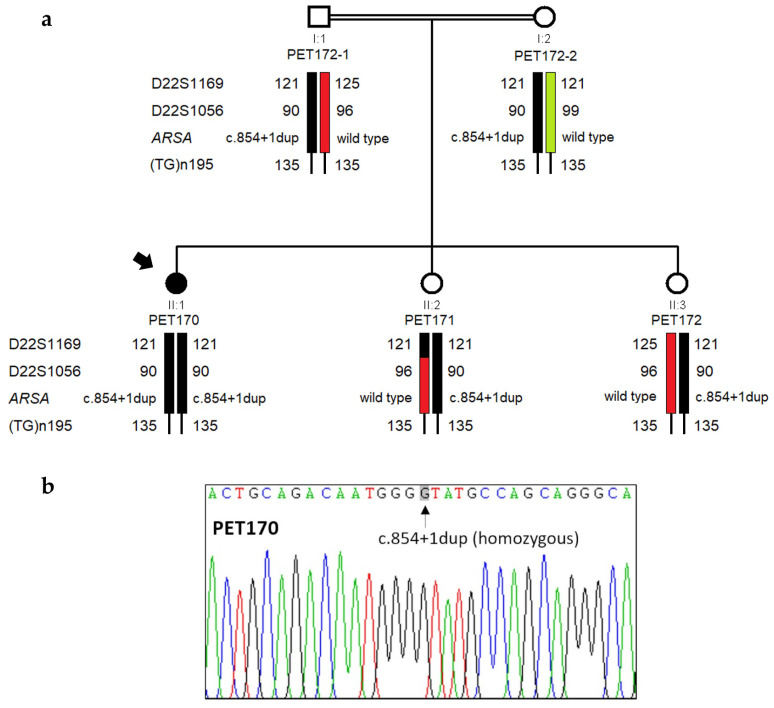
Genetic analysis of family 1. (**a**) Pedigree of family 1, showing chromosomal segregation with microsatellite alleles and ARSA gene variants. The distal marker (TG)_n_195 is uninformative in this family. (**b**) Electropherogram showing the homozygosity of the novel mutation c.854+1dup in the proband (PET170).

**Figure 3 genes-15-01515-f003:**
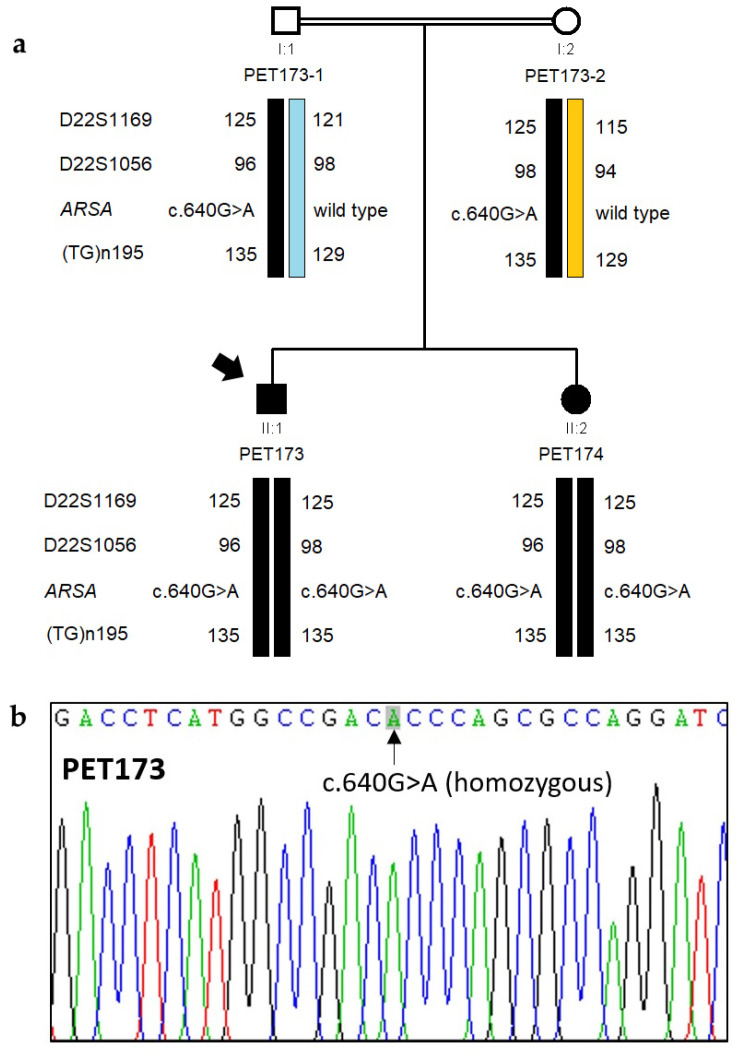
Genetic analysis of family 2. (**a**). Pedigree of family 2, showing chromosomal segregation with microsatellite alleles and ARSA gene variant c.640G>A (**b**) Electropherogram showing the mutation c.640G>A in homozygosity in the proband (PET173).

**Figure 4 genes-15-01515-f004:**
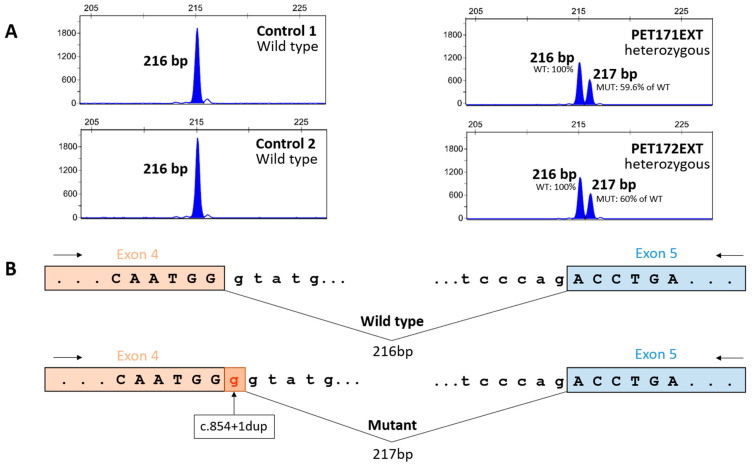
Pathogenic effect of the *ARSA* variant c.854+1dup. (**A**) Capillary electrophoresis of cDNA amplification products from exon 4 to exon 5 of the *ARSA* gene in two control individuals and two siblings heterozygous for the c.854+1dup variant. We obtained an amplicon of the expected size (216 bp) in wild-type individuals, while the carrier siblings showed one amplicon of the expected size and another with an extra base pair (217 bp). Areas under the curve are equivalent to the amount of amplicon detected, with the amount of the c.854+1dup allele (PET171: 3572 fluorescent units (FU); PET172: 3597 FU) being about 60% of the wild-type allele (PET171: 5992 FU; PET172: 5995 FU). (**B**) Schematic representation of the results in panel (**A**), showing that splicing of the mutant allele adds an extra base to exon 4 (as verified by Sanger sequencing of the amplicons), leading to a frameshift and the appearance of a premature termination codon.

**Table 1 genes-15-01515-t001:** Primers and PCR conditions for amplicons containing causative variants.

Target ^1^	Exons	Primer Sequences (5′-3′)	Annealing T (°C)	Amplicon (bp)
gDNA, *ARSA*	3–4	Upper: CCATCGATTTCTAGGCATCCC	64	708
		Lower: TCACCCACTATGTTCTTGGCAA		
cDNA, *ARSA*	4–5	Upper: **FAM**- CAGAGCTTTGCAGAGCGTTCA	64	216
		Lower: CTCGTAGGTCGTTCCCTTTCCA		
gDNA, *GNPTAB*	13 (part)	Upper: CTCAGACTCAAAGAATTAAAGGAA	55	536
		Lower: GGGCTCTCCTTGTTGAGTTA		
gDNA, *MANBA*	14	Upper: CTTCTCTCATGCTAAGGGGCTAGT	58	498
		Lower: GAGTTGGGTGGCTGTAGTTCC		

^1^ gDNA: genomic DNA. cDNA: DNA copy of an mRNA molecule.

**Table 2 genes-15-01515-t002:** Microsatellite markers flanking the *ARSA* gene used in this work.

Name	Primer Sequences (5′-3′)	Annealing T (°C)	Amplicon (bp)
D22S1169	Upper: **FAM**- GCACACACATGCACATAATC	56	118–138
	Lower: AACAACTTCCAGCAGACG		
D22S1056	Upper: **HEX**- CACCCCCCCAAAAAAGTGT	56	90–100
	Lower: ATGCTGTTTCTCACCCCAGT		
(TG)_n_195	Upper: **HEX**- AAACGTTTGACTGAGCCAAGCA	56	~135
	Lower: GTCCACTCACCCACGCACAGA		

**Table 3 genes-15-01515-t003:** Assessment of pathogenicity of the variants reported in this work.

Gene ^1^	Variant		Minor Allele Frequency (MAF) ^2^	ACMG Criteria	Classification	Reference
	DNA	Protein				
*ARSA*	c.640G>A	p.(Ala214Thr)	4 × 10^−6^ (global) 1 × 10^−5^ (African)	PM1, PM2, PM5	Likely pathogenic	[29]
*ARSA*	c.854+1dup	p.(Pro286Thr*fs2)	0 (global)	PVS1, PM2, PP3	Pathogenic	This work
*GNPTAB*	c.1931_1932inv	p.(Thr644Met)	0 (global)	PS3, PS4, PM2, PM3	Pathogenic	[30]
*MANBA*	c.1922G>A	p.(Arg641His)	7 × 10^−5^ (global) 4 × 10^−5^ (African)	PS3, PS4, PM2, PP3	Pathogenic	[31]

^1^ Accession numbers: *ARSA*, NM_000487.6; *GNPTAB*, NM_024312.5; *MANBA*, NM_005908.4. ^2^ From gnomAD 4.1.0.

## Data Availability

Data on the *ARSA* pathogenic variants that we report in this study, c.640G>A and c.854+1dup, are available in the ClinVar database (accession numbers SCV005367821 and SCV005368674, respectively) [22].

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
