# Peer review of "Metachromatic Leukodystrophy in Morocco: Identification of Causative Variants by Next-Generation Sequencing (NGS)"

_genes, 2024, doi:10.3390/genes15121515_

Round 1
Reviewer 1 Report
Comments and Suggestions for Authors
I have read with care the article written by Hammoud and collaborators. Overall, the article summarizes technics and managements that have become routine in most Western countries. They describe obtaining molecular diagnoses in relatively classical disorders, with very common tests and technics. The fact that they identify new variants is not sufficient to justify publication of these data. Databases such as clinvar are made for this purpose.
Although the article is of overall great quality, it doesn’t provide any new information for the scientific community.
Minor remarks :
- It is a shame that the brain MRI of these patients are not shown, as MLD is above all a leukodystrophy
- Genetic analyses are done in a gene panel approach
- The authors briefly mention that one individuals also had variants in other genes associated with inherited metabolic disorders, including some which clinical phenotype overlaps with MLD. Unfortunately the analysis of the impact of this discovery is not further developed.
Reviewer 2 Report
Comments and Suggestions for Authors
Dear Authors,
Thank you for the opportunity to review your manuscript on MLD genetic diagnosis in Moroccan patients. Your work addresses an important topic in rare disease diagnostics, and the identification of novel variants contributes valuable information to the field. However, I have several concerns that should be addressed to strengthen the manuscript.
Major Comments:
1. Patient Cohort and Clinical Data:
- Please clarify your selection criteria from the initial 73 patients to the final 3 cases
- Expand the clinical descriptions, particularly:
* Detailed neurological symptoms progression
* Quantitative MRI findings with images if possible
* Complete biochemical profiles with reference ranges
- A timeline diagram of symptom onset and progression would be helpful
2. Methodological Clarifications:
- For biochemical analyses:
* Include validation data for your methods
* Provide QC criteria
* Add technical replicates information
- For NGS analysis:
* Clarify variant filtering criteria
* Explain how you determined coverage requirements
* Detail your variant validation strategy
3. Results Enhancement:
- The functional impact of the novel c.854+1dup variant:
* Add quantitative data from splicing assays
* Include protein expression analysis if available
- For the c.640G>A variant:
* Provide additional evidence supporting pathogenicity
* Consider functional studies
- GNPTAB variant impact:
* Strengthen the connection to phenotype modification
* Consider additional functional evidence
4. Discussion Development:
- Expand on:
* Intrafamilial phenotypic variations
* Comparison with other population studies
* Clinical implications for Moroccan healthcare
* Screening recommendations
Minor Comments:
1. Consider adding a table summarizing:
- Clinical features
- Biochemical results
- Genetic findings
- Phenotype-genotype correlations
2. For the methods section:
- Add detailed protocol steps
- Include reagent sources
- Provide analysis parameters
3. For the discussion:
- Address diagnostic challenges in resource-limited settings
- Discuss screening implementation possibilities
- Consider cost-effectiveness aspects
Your study has the potential to make a significant contribution to the field. I believe addressing these points will strengthen your manuscript considerably. Please feel free to contact me through the editor if you need any clarification on these comments.
Best regards,
Reviewer

The English could be improved to more clearly express the research.
Round 2
Reviewer 1 Report
Comments and Suggestions for Authors
In this revised manuscript :
- the authors have adressed each concerns that were raised by me and the other reviewers
- the authors added the description of the brain MRIs as images were not available. This lengthens the manusciript but remains of interest i think.
My main concern remains the overall interest of the article, but this concern is not shared by other reviewers and the work remains of overall great quality
Author Response
Dear Editor:
We have carefully read the new reports from both reviewers. We are grateful for all the advice to improve our manuscript.
Please find below a point-by-point response to every comment raised by the two reviewers. All the changes that we implemented in the manuscript appear highlighted in yellow.
This third version has also been revised by a professional service for scientific revision in English, as suggested by Reviewer 2, in addition to the native English speaker that revised the previous version.
Best regards,
Dr Francisco J. del Castillo (corresponding author, on behalf of all authors)
Reviewer 1
In this revised manuscript :
- the authors have adressed each concerns that were raised by me and the other reviewers
- the authors added the description of the brain MRIs as images were not available. This lengthens the manusciript but remains of interest i think.
My main concern remains the overall interest of the article, but this concern is not shared by other reviewers and the work remains of overall great quality
We thank very much Reviewer 1 for the high opinion of our manuscript and the endorsement of our work.

Reviewer 2 Report
Comments and Suggestions for Authors
Dear Authors,
Thank you for the opportunity to review your revised manuscript. I appreciate the thorough responses to the previous reviewers' comments and the improvements made to the manuscript. While the core scientific content is now strong, I have a few minor suggestions that could further enhance the clarity and completeness of your work:
1. Figures and Visual Presentation:
- Consider adding a simplified pedigree diagram combining both families in the abstract or early in the paper to give readers an immediate overview of the cases
- In Figure 3, consider adding percentages directly on the electropherogram peaks for easier comparison of wild-type vs mutant transcript levels
- A schematic timeline showing the typical progression of MLD symptoms versus the onset observed in your cases would be helpful for clinical context
2. Discussion Enhancement:
- Brief discussion on how your findings could impact genetic counseling approaches for MLD in consanguineous populations
- Consider adding a short paragraph on cost considerations for implementing your diagnostic approach in resource-limited settings
- Add a note on potential limitations of your splicing assay methodology
These minor revisions would improve the manuscript's clarity and completeness. The fundamental scientific content is sound and these suggestions are mainly to enhance presentation and accessibility for readers.
Please feel free to contact me through the editor if you need any clarification on these comments.
Best regards,
Comments on the Quality of English Language
The English could be improved to more clearly express the research.
Author Response
Dear Editor:
We have carefully read the new reports from both reviewers. We are grateful for all the advice to improve our manuscript.
Please find below a point-by-point response to every comment raised by the two reviewers. All the changes that we implemented in the manuscript appear highlighted in yellow.
This third version has also been revised by a professional service for scientific revision in English, as suggested by Reviewer 2, in addition to the native English speaker that revised the previous version.
Best regards,
Dr Francisco J. del Castillo (corresponding author, on behalf of all authors)
Reviewer 2
Dear Authors,
Thank you for the opportunity to review your revised manuscript. I appreciate the thorough responses to the previous reviewers' comments and the improvements made to the manuscript. While the core scientific content is now strong, I have a few minor suggestions that could further enhance the clarity and completeness of your work:
- Figures and Visual Presentation:
- Consider adding a simplified pedigree diagram combining both families in the abstract or early in the paper to give readers an immediate overview of the cases
Done as suggested, it is now Figure 1.
- In Figure 3, consider adding percentages directly on the electropherogram peaks for easier comparison of wild-type vs mutant transcript levels
Done as suggested.
- A schematic timeline showing the typical progression of MLD symptoms versus the onset observed in your cases would be helpful for clinical context
Done as suggested, it appears as Figure S1 within the Supplementary Information.
- Discussion Enhancement:
- Brief discussion on how your findings could impact genetic counseling approaches for MLD in consanguineous populations
Added as suggested in the discussion (lines 433-436).
- Consider adding a short paragraph on cost considerations for implementing your diagnostic approach in resource-limited settings
We had already discussed this briefly. We have stressed this topic now in lines 436-441.
- Add a note on potential limitations of your splicing assay methodology
Added as suggested (lines 369-373).
These minor revisions would improve the manuscript's clarity and completeness. The fundamental scientific content is sound and these suggestions are mainly to enhance presentation and accessibility for readers.
Please feel free to contact me through the editor if you need any clarification on these comments.
Best regards,
